# Evaluation of Antimicrobial Usage Supply Chain and Monitoring in the Livestock and Poultry Sector of Pakistan

**DOI:** 10.3390/vetsci12030215

**Published:** 2025-03-01

**Authors:** Muhammad Farooq Tahir, Riasat Wasee Ullah, Jing Wang, Kinzang Dukpa, Muhammad Usman Zaheer, Sami Ullah Khan Bahadur, Usman Talib, Javaria Alam, Muhammad Akram, Mo Salman, Hamid Irshad

**Affiliations:** 1Integral Global, Atlanta, GA 30084, USA; 2The Fleming Fund Country Grant Pakistan, Health Security Partners, Islamabad 44000, Pakistan; usman.zaheer@fao.org (M.U.Z.); sami.bahadur@colostate.edu (S.U.K.B.); avalam@ucdavis.edu (J.A.); 3Livestock Wing, Ministry of National Food Security and Research, Islamabad 44000, Pakistan; riasat.waseeullah@fao.org (R.W.U.); ahc@monfsr.gov.pk (M.A.); 4World Organisation for Animal Health Regional Representation for Asia and the Pacific, Tokyo 113-0032, Japan; j.wang@iaea.org (J.W.); k.dukpa@woah.org (K.D.); 5Regional Office for Asia and the Pacific, Food and Agriculture Organization of the United Nations, Bangkok 10200, Thailand; 6The Fleming Fund Country Grant Pakistan, DAI, Islamabad 44000, Pakistan; usman_talib@dai.com; 7Animal Population Health Institute, College of Veterinary Medicine and Biomedical Sciences, Colorado State University, Campus Stop 1644, Fort Collins, CO 80523, USA; mo.salman@colostate.edu; 8Animal Health Program, Animal Sciences Institute, National Agricultural Research Centre, Islamabad 44000, Pakistan; hamid@parc.gov.pk

**Keywords:** AMR, AMU, antimicrobials, regulation, supply chain, surveillance, livestock, poultry

## Abstract

The overuse and misuse of antimicrobials is considered a major factor in the development of antimicrobial resistance (AMR). Monitoring the use of antimicrobials is an effective tool to control AMR. However, there is no systematic antimicrobial use (AMU) monitoring in the livestock sector of Pakistan, which may result in its unregulated use. This paper discusses the overall status of the antimicrobial supply chain in the livestock sector of Pakistan. It presents results from a systems mapping exercise of key stakeholders with aims to map the supply chain of antimicrobials from import and manufacturing to farm use, identify gaps in its regulation, and develop recommendations for improvement. The AMU supply chain was mapped visually to illustrate the key players and their roles at each stage. Major challenges identified were a fragmented AMU supply chain co-regulated by various stakeholders, weak regulatory framework, and the lack of comprehensive database. Recommendations included strengthening legislation, restricting the use of antibiotics as growth promoters, developing a national AMU database, and raising stakeholder awareness of the responsible use of antimicrobials. This study provides a model for low-resource settings facing similar AMR challenges on how to map the antimicrobial supply chain, identify gaps, and improve AMU surveillance to combat AMR.

## 1. Introduction

In the recent decades, the world has witnessed an increasing emergence of antimicrobial resistance (AMR), primarily due to misuse and overuse of antimicrobials in both humans and animals [1]. AMR is being considered a silent pandemic. With many countries in the world facing serious public health crises due to antimicrobial resistance (AMR), it is crucial to explore the root causes of the problem [2]. This exploration is necessary for implementing preventive measures both nationally and globally. Although the causes of AMR development are well known, there are limited data on various associated factors in middle- and low-income countries. Basic information, supported by reliable data, regarding the use and misuse of antibiotics in human and veterinary medicine, as well as in plants, is either limited or absent. Without such information, it is challenging to determine the causes of AMR [3].

Pakistan, with its available scientific resources, funding opportunities, and a deep interest in the subject, can be considered a model for other similar countries in investigating the causes of AMR.

Pakistan has a large rural and agriculture-based industry, and animal husbandry plays an important role in the economy of Pakistan and is a major source of livelihood for many farmers. Livestock plays a significant role in the socio-economic development of the country. In the year 2022–23, the livestock sector contributed 14.3% of the Gross Domestic Product (GDP) to the national economy. The livestock sector provides employment to 30 million people, especially in rural areas [4]. The increasing demand for livestock due to urbanization, an increase in population, and export opportunities leads to the increasing trend of intensive farming in Pakistan. Therefore, farmers are extensively using antibiotics in food-producing animals as growth promoters and for treatment and prophylactic purposes. A recent study carried out in Pakistan to assess the knowledge, attitudes, and practices (KAP) of field veterinarians regarding AMR and antimicrobial use (AMU) indicated an irrational prescription of antibiotics, including for treatment of viral diseases and prophylactic purposes [5]. Most studies have shown increasing rates of resistance in both humans and animals, with multi-drug-resistant organisms (MDROs) being isolated with increasing frequency across the country. The use of antimicrobial agents in animals, poultry, and agriculture possesses recognized benefits, but overuse has potentially serious implications for human health. Antimicrobial Stewardship Programs (ASPs), Infection Control Programs, and AMR/AMU monitoring and surveillance are the most effective immediate options for combatting and monitoring the trend of AMU and, subsequently, AMR. As such, understanding the access to antimicrobials and mapping the supply chain is inevitable to identify areas of interventions for an effective ASP, in addition to other actions around AMU monitoring at farm-level and AMR surveillance in animals. Unfortunately, in Pakistan, the supply chain of antimicrobials has not been mapped yet, which limits the country’s ability to develop and institutionalize robust ASPs.

Pakistan has developed a national AMR surveillance strategy for healthy and diseased food animals (cattle, buffalo, and poultry). A surveillance network comprising two National Reference laboratories i.e., the National Reference Laboratory for Poultry Diseases (NRLPD) and the National Veterinary Laboratory (NVL), and nine peripheral laboratories has been established [6]. Due to inefficient ASPs and extensive use of antibiotics in Pakistan, it is important to monitor the quantities and usage patterns of antimicrobials in animals by analyzing the whole AMU supply chain in terms of registration, procurement, production, distribution, management, and monitoring of usage of antimicrobials in Pakistan. The Ministry of National Food Security and Research (MoNFS&R) has been trying to share reliable annual data on antimicrobial usage in Pakistan with the World Organisation for Animal Health (WOAH). However, there are still challenges faced in the collection of quality and comprehensive data from various sources, including the livestock and aquaculture industries, government authorities, and other private sectors [7]. The information on AMU in animals is crucial for identifying potential risk factors and enabling effective control and prevention of AMR in animals and, subsequently, in humans [8]. This requires a comprehensive surveillance system to monitor the use and fate of antimicrobials in animal diseases in Pakistan. However, before implementing such a surveillance system, it is essential to gather basic information and understand the steps in the flow of antimicrobials from manufacturing to administration as well as their presence in animal-origin food items. A functional national monitoring system for antimicrobials used in animals can provide valuable information about the type and quantity of antimicrobials used, and this information can be correlated with the resistance patterns to provide policy decisions.

The MoNFS&R, owing to its interest in tackling AMR in food animals, with the support of WOAH and Fleming Fund, organized a workshop in March 2020 on the monitoring of AMU in the animal health sector in Pakistan. The key objectives were to map the antimicrobial supply chain and support establishment of a system for monitoring the quantities and usage patterns of antimicrobial agents used in the livestock sector in Pakistan. The aim of this paper is to present a study conducted to map and understand the antimicrobial supply chain in the livestock and poultry sector through inputs from stakeholders gathered for this purpose. This information will assist in designing and initiating a monitoring and surveillance system for AMU among various segments of the livestock sectors in Pakistan.

## 2. Materials and Methods

### 2.1. Methodological Approach

A systems thinking approach was adopted for the analysis of the stakeholders engaged in the AMU supply chain [9]. The approach was modified as per requirements, and a mix of network analysis and focused group-discussion methods were adopted. The process was divided into three main steps: identification of the key stakeholders, data collection from the stakeholders, and network analysis of the stakeholders. Chief Veterinary Office/Livestock Wing of the Ministry of National Food Security and Research led the process with technical and financial assistance from the WOAH Regional Representation Office in Japan and the Fleming Fund Country Grant, Pakistan.

### 2.2. Selection Criteria for Stakeholders:

To ensure diverse representation of participants from both public and private sectors for the AMU monitoring workshop, the following selection criteria were considered:(a)Institutional Representation: Stakeholders or key institutions engaged in import, regulation, distribution, and use of antimicrobials, such as regulatory bodies, the human health sector, veterinary departments, Customs, veterinary educational establishments, fisheries and aquaculture, drug quality testing laboratories, and research institutes.(b)Expertise and Knowledge: Participants with expertise in veterinary medicine, animal husbandry, pharmaceuticals, public health, and regulatory affairs related to antimicrobials.(c)Geographical Representation: Participation from all regions or provinces within the country to capture regional variations in AMU practices and regulations.

Based on the set criteria, international, federal, and provincial authorities, organizations, and stakeholders, as given Section 2.3, were identified and invited to the workshop.

### 2.3. Stakeholder Enagaged

Based on the selection criteria, the following stakeholders were engaged for the discussions during the workshop.

(a)International Organizations: International organizations invited to the workshop included the World Organisation for Animal Health, World Health Organization, Pakistan (WHO), Food and Agriculture Organization, Pakistan (FAO), and the Fleming Fund Country Grant, Pakistan.(b)National Institutions Working with Import of Antimicrobials: These included the Ministry of National Food Security and Research, Drug Regulatory Authority of Pakistan (DRAP), Customs Department, Pakistan, and poultry, dairy and pharmaceutical associations.(c)National Institutions Working with Regulation of Antimicrobials: In addition to the DRAP, it included National Veterinary Laboratories, Islamabad.(d)National Institutions Working with Use of Antimicrobials: These included the Fisheries Development Board, Livestock and Dairy Development Board, and Directorate of Livestock, Islamabad, on the federal level. Livestock and Dairy Development Departments (L&DD) from Punjab, Khyber Pakhtunkhaw, Sindh, Balochistan, Azad Jammu and Kashmir, and Gilgit Baltistan represented the provincial and regional governments. University of Agriculture, Faisalabad, and Islamia University, Bahalwapur, represented the veterinary academic institutions. The Poultry, Dairy, and Veterinary Pharmaceutical Associations participated as private sector representatives.

### 2.4. Development of Activity Sheets and Data Collection

Activity sheets were developed prior to the workshop to gather information on the AMU supply chain and monitoring. The participants were asked to describe the status of AMU supply chain in their respective provinces or regions and their organization’s role in a range of topics. These included availability of AMU monitoring policy or regulations, if the existing legislations are sufficient for AMU monitoring in livestock sectors, and the implementation status of these legislations. They were also asked to identify the gaps and point out action items to bridge these gaps. These activity sheets were provided to the nominated participants prior to the workshop, and the same sheets were used in the focused group discussions during the workshop.

### 2.5. Stakeholder Consultation/Analysis

(a)Mapping of AMU Supply Chain: During the first day of the workshop, regulatory authorities and other participants were divided into groups to map the sources of the supply chain of antimicrobial agents intended for use in animals. However, before seeking this important information, the objectives of the activity were discussed. To sensitize the participants and emphasize the significance of national data reporting and regulatory procedures at federal and provincial levels, detailed presentations were delivered. These covered topics such as the global AMU database and the importance of national data reporting (presented by WOAH), the regulatory process around antimicrobial import, registration, sale, and usage (presented by the Drug Regulatory Authority of Pakistan), the regulatory frameworks for monitoring antimicrobial usage (presented by the Livestock and Dairy Development Department Punjab), and the import process and real-time surveillance capabilities for antimicrobials (presented by the Customs Department). This preparation aimed to ensure effective group discussions focused on mapping the AMU supply chain.(b)AMU Monitoring: On second day of the workshop, following the technical presentations on status and activities on AMR and AMU by various stakeholders, the participants were divided into three groups based on their expertise to brainstorm on (i) current legislation and governance status for monitoring of AMU in Pakistan to identify the key government regulators and their roles in regulating AMU along the supply chain; (ii) national AMU data collection and monitoring system to identify types of AMU data that each stakeholder can provide to the MoNFS&R, and (iii) monitoring of AMU at end-user level to identify priority activities to enhance national AMU-monitoring system in Pakistan. The aim of this exercise was to consider the various steps and conditions for marketing and using various antimicrobial products among livestock species in Pakistan. Prior to the workshop, the organizers drafted a map of antimicrobial chain supplies and shared it with the breakout groups to confirm and modify the pathways for AMU in livestock species.

Result compilation and concurrence: At the end of each exercise, every group presented their findings, and based on that, the drafted results were developed with the consensus of the participants.

## 3. Results

### 3.1. The AMU Supply Chain of Pakistan

Based on the findings and/or inputs of all groups, a flow diagram (Figure 1) on antimicrobial supply chain in Pakistan was developed with the consensus of the participants. Presently, the drug regulatory authority of Pakistan (DRAP) of the federal government regulates the import, manufacture, registration, and pricing of antimicrobials, while provincial governments regulate distribution, sale, and post-marketing inspection (jointly with DRAP) in the provinces. The Pakistan Customs Department deals with the clearance of import and export of drugs. Pakistan Customs have strong real-time surveillance system, and therefore the MoNFS&R can access the data bank of the Customs Department for data on antimicrobials and active pharmaceutical ingredients (APIs) imported into Pakistan. The provincial livestock departments regulate animal production systems, prescriptions, and withdrawal observations in livestock farms and clinics.

Only pharmaceutical companies are allowed to import APIs, whereas finished pharmaceutical products (FPPs) can be imported by pharmaceutical companies and importers. As per the AMU supply chain, the importers import FPPs and sell to wholesalers, retailers, the feed industry, and sometimes even to farms directly. Feed industries import feed premix containing antibiotics and purchase FPPs from importers to be added to animal feed based on demand. Medicated feed (containing antibiotics) and FPPs are then distributed to retailers, veterinary pharmacies, and veterinary service centers through distributors. Antimicrobials are ultimately used by farms, veterinarians, para-veterinary staff, veterinary hospitals/clinic, zoos, and aquaculture. As for AMU data, presently, only higher-level AMU data are available, such as from importers (import data), customs offices (customs data), manufacturers (manufacturing at a higher level), and wholesalers (sales data). Currently, no AMU data at the lower level of the AMU supply chain are available, such as retailer/pharmacy sale data, farm level-usage data, and prescription data from clinics and veterinary hospitals, etc.

Currently, at the federal level, there are no regulations on import, manufacture, registration, and sale of animal feed premix (medicated feed) containing antimicrobials. However, some provinces such as Punjab have legislations regulating use of medicated poultry feed and also have lists of permissible and non-permissible antimicrobials used in poultry feed. According to the Punjab Poultry Production Act-2016 [10], antimicrobials such as colistin and ciprofloxacin are banned for use as antimicrobial growth promoter in poultry feed. The government wants to take a cautious approach in banning the use of antimicrobial growth promotors (AGPs) in poultry production and will develop step wise approach in phasing out the use of antimicrobials for growth promotion.

### 3.2. Legislation and Governance Status for Monitoring of AMU in Pakistan

There is currently no policy or legislation system to monitor the use of antimicrobials at the national and provincial level. There is a need to review and update the existing DRAP Act by adding relevant clauses to accommodate the requirement for regulations of veterinary medicines including antimicrobials. This should be led by DRAP and Ministry of Health MoNHSRC with the collaboration of MoNFS&R. Likewise, the revised national legislations should cover reporting of antimicrobial usage at prescriber and farm level and should also include monitoring of premixes containing antimicrobials by the provincial livestock departments in collaboration with MoNFS&R and DRAP. Regulations of unauthorized usage of antimicrobials need to be developed by reviewing the existing legislations by provincial health care commissions and the Pakistan Veterinary Medical Council involving provincial livestock departments, including developing a legal framework for prescription requirements for the veterinary practitioners. The legislations related to acceptable daily intake and maximum residue levels are already available but need to be implemented through provincial livestock departments involving DRAP and MoNFS&R.

### 3.3. National AMU Data Collection and Monitoring System

Table 1 shows the types of AMU data that each stakeholder can provide to the Ministry of National Food Security and Research. Currently, Pakistan does not have a national AMU data collection and monitoring system for antimicrobials used in animals. Therefore, it is important to develop a national AMU monitoring system to be led by the Animal Husbandry Commissioner’s office (AHC) of MoNFS&R jointly with the Fisheries Development Commissioner (FDC),the Ministry of Maritime Affairs (MoMA) in collaboration with Drug Regulatory Authority of Pakistan, Pakistan Customs Department, and provincial livestock departments. It is important to make a list of critical antimicrobials that are of importance for Pakistan based on WHO’s list of critical antimicrobials and WOAH’s list of antimicrobial agents of veterinary importance.

The AHC office and FDC should also make a list of importers of antimicrobials with the support of DRAP and the Pakistan Customs Department that could be contacted for the submission of import data. The AHC and FDC office should also develop necessary guidelines for monitoring of antimicrobials used in animals and share them with enforcement agencies such as DRAP, Pakistan Customs, provincial livestock departments, provincial drug regulation authorities, and health departments. The national AMU-monitoring system should clearly spell out the roles and responsibilities of relevant stakeholders identified in the AMU supply chain, particularly for their roles in collection and sharing of AMU data with the MoNFS&R. A national AMU database system should be developed that will also be the repository for AMU data from the animal sector. It is important to advocate awareness of the importance of AMU monitoring so that AMU data collection can be harmonized at all levels of the government and the private sector.

### 3.4. Monitoring of AMU at End-User Level

Table 2 shows the status of AMU monitoring and data collection at end user level, identified gaps and way forward for establishing a robust AMU monitoring system.

AMU Monitoring: According to identified priority activities to enhance the national AMU monitoring system, Pakistan lacks an established system for monitoring AMU. To address this, a comprehensive surveillance plan and legislation were proposed, involving collaboration between federal and provincial entities as well as both public and private sectors.

Guidelines/SOPs: Similarly, no guidelines and standard operating procedures (SOPs) for AMU monitoring exist. The forward strategy should involve collaborative efforts from federal and provincial authorities with academia, livestock departments, research institutes, associations, and fish departments.

Development of a monitoring system: Currently, there is a lack of a centralized database for AMU monitoring. The proposed way forward includes initiating a pilot study in various production systems, which requires collaboration between federal and provincial bodies, as well as involving livestock departments and associations.

Publications: Due to limited published data, proposed plans include conducting more studies in collaboration with academia, research institutes, livestock departments, and associations.

Implementations and Challenges: Several challenges, such as a lack of awareness, trained manpower, funding, and political interest, were recognized. To address these challenges, budget allocation, political will, and training were identified as impactful solutions. This comprehensive approach requires collaboration between federal and provincial entities and planning and finance departments.

Roles and responsibilities: Clarification and establishment of roles and responsibilities are essential, focusing on federal legislation. This requires legislation, implementation, data analysis, and reporting.

Knowledge, Attitudes, and Practices surveys: In 2020, there was no planned activity for KAP (Knowledge, Attitudes, and Practices) surveys. The suggested plans include a comprehensive study with a multi-sectorial approach, raising awareness, and observing AMR awareness week. These activities should be led by livestock departments, with the collaboration of academia, research institutes, Non-Governmental Organizations (NGOs), associations, and farmers.

## 4. Discussion

Pakistan imports an abundant quantity of antimicrobials for human and animal sectors. The value of imports of antibiotics as a commodity to Pakistan was $127 million in 2020, which indicated a 5.8% increase compared to 2019 [12]. Monitoring animal antimicrobial usage is crucial to curb irrational use. Currently, DRAP oversees the federal regulation of antimicrobial import, manufacture, registration, and pricing for animals, while provinces, jointly with DRAP, regulate distribution, sale, and post-marketing inspection. Pakistan lacks a national monitoring system for animal antimicrobial usage. WOAH recommends implementing such a system to prevent misuse and ensure access to quality veterinary medicines [13]. Therefore, regulatory authorities can use findings of the current study to make informed decisions regarding the regulation, import, and manufacture of antimicrobials for animals. By identifying gaps in existing legislation and governance related to AMU, this study suggests a roadmap for regulatory improvements. Similarly, suggested national AMU data collection and monitoring systems can help track and manage antimicrobial use, preventing misuse and overuse.

Overall, Pakistan’s veterinary drugs supply chain is fragmented. At a national level, it is regulated by the DRAP Pakistan, which does not have in-house veterinary specific expertise and coordinates with the CVO office for veterinary specific consultations. This is a major gap which results in certain regulatory discrepancies, e.g., the veterinary feed additives are not regulated by the DRAP; however, a significant amount of antibiotics are being imported in the country under the label of growth promoters. This also leads to inconsistent data of the veterinary antimicrobial imports and usage in the country. There is a lack of veterinarian–famer interaction, which further aggravates the irrational use of antibiotics, which is very similar to the veterinary antimicrobials’ supply chain issues in Lao [14]. Certain other factors also impede the collection of quantitative data on AMU, including ‘lack of regulatory framework’, ‘lack of coordination between national authorities and private sector’, ‘lack of tools and human resources’, and ‘insufficient regulatory enforcement’ [15]. The countries in Europe have led from the front in effective AMU monitoring in the animal sector that have resulted in reduction of AMR over the years. For instance, Denmark has developed the Danish Integrated Antimicrobial Resistance Monitoring and Research Programme (DANMAP) using one health approach, which not only collects AMU data from both human and animal sectors but also monitors the AMR. This program works through the collaboration of the Ministry of Health, the Ministry of Higher Education and Science, and the Ministry of Environment and Food [16]. Denmark is also a part of European Surveillance of Veterinary Antimicrobial Consumption (ESVAC), which was launched in 2009 for the harmonization of antimicrobial usage data in some European countries. For this purpose, they have developed the method for standardizing antimicrobial sales and designed the format to collect data from all member countries [17]. This approach can be implemented at the national level in countries like Pakistan, involving all provinces/states. A drug-distribution framework has been decided and strictly followed in the USA [18], which makes it easier to collect data on AMC and monitor the antimicrobials in the country. In parallel, the USA has the Animal Health Institute as a trade association of manufacturers of pharmaceuticals and animal health care products that releases an AMU/AMC estimate every year. This association also reports the total active ingredients manufactured and used in growth promotors. The US Food and Drug Authority (FDA), in collaboration with the animal health industry, has also updated the guidelines for the industry to make changes in antibiotic labels to ensure the judicious use of medically important antibiotics.

A functional national monitoring system for AMU in animals can provide valuable information about the type and quantity of antimicrobials used, and this information can be correlated with the resistance patterns to make policy decisions. Such a functional AMU monitoring system can help optimize the use of antibiotics in the low- and middle-income countries (LMICs) [19]. Therefore, there is a need to develop an integrated national AMU database to integrate data collected from all the private sectors/companies/importers/distributors and end users dealing with antimicrobials. Based on the outputs of this workshop and the AMU data available with the end users in Pakistan, a potential flow sheet for a Point Prevalence Survey (PPS) for the estimation of AMU was designed. Following this, PPS surveys have been conducted in commercial broilers [20] and dairy production of Pakistan. These surveys were designed in line with the recommendations of the supply chain mapping exercise and provide high-resolution data on the farm-level use of antimicrobials (class, quantity, mode of administration, etc.), including seasonal and geographic differences in AMU. The findings of these studies clearly indicate the need to institute a robust national AMU monitoring system in Pakistan involving the key stakeholders at all levels. Based on the findings of activity on monitoring of AMU at end user level, Knowledge, Attitude, and Practices (KAPs) surveys on AMU for field veterinarians and farmers were designed and conducted at a national level, covering all geographical areas of Pakistan. Another highly significant consequence of this activity was the collection of antimicrobial imported data from the Drug and Regulatory Authority of Pakistan, which was subsequently provided to World Organisation for Animal Health (WOAH) as per the described format by WOAH. Sharing insights and data on AMU in the livestock sector enhances Pakistan’s participation in global initiatives addressing AMR. It fosters collaboration with international organizations, promoting a unified response to a shared public health threat. Therefore, activities under this study are highly important as a direction for other countries. This study also highlights the need for improved legislations and enhanced enforcement of regulations to ensure prudent use of antimicrobials in the livestock sector.

## 5. Conclusions

Considering the availability of financial resources and data, developing a national level AMU data collection system from sales and import data according to the WOAH methodology can be considered as a starting point for Pakistan, while in parallel, exploring details of the AMU situation by using PPS. The development of a national AMU monitoring system in Pakistan will help us understand the current situation of national monitoring of AMU in Pakistan in livestock and poultry sectors, including the legislation and governance system, trends in the import of antimicrobials, the status of AMU data monitoring (including AMU data collection, analysis, publication, and sharing within and outside Pakistan), and the roles and responsibilities of stakeholders in AMU data sharing. This includes other sectors such as customs, agriculture, importers and distributors/retailers (medical pharmacies, feed agents), and private farms, including those importing informally.

## 6. Recommendations

Key recommendations of this study include strengthening legislation from antimicrobial import and manufacturing until its use at farms. Pakistan needs to restrict the use of antibiotics as growth promoters in animal health sectors, especially poultry. The country needs to establish a comprehensive national database on AMU and raise awareness of stakeholders on the responsible use of antimicrobials.

## Figures and Tables

**Figure 1 vetsci-12-00215-f001:**
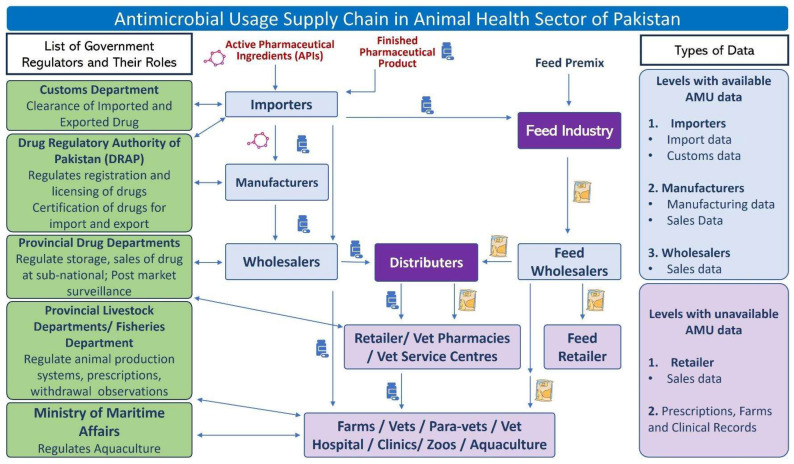
Existing antimicrobial supply chain in veterinary sector of Pakistan.

**Table 1 vetsci-12-00215-t001:** Current status, gaps, and collaborative solutions to improve national AMU data collection system.

Topic	Current Gaps/Challenges	Way Forward	Lead Implementer	Collaborator	Timeline
National Data Collection System	No system in place.	AMU data collection system to be developed	AHC MoNFS&R FDC, MoMA	DRAP, Customs, Prov LDD, Prov Drug Regulatory Authorities (DRAs)	
Type of AMU Data	No specific demarcation on Critical AMs/GPs, etc.	Differentiation of AMs according to intl. standards	AHC, FDC	DRAP, Customs, Prov LDD, Prov DRA	Short term
Main Importers of AMs	Most importers are private pharmaceutical companies and importers/distributors.	List of importers needs to be identified and recorded.	AHC and FDC	DRAP, Customs	Short term
List of AMs Harmonized as per WOAH and WHO List	No list of AMs identified and notified as per critical AMs notified by WOAH and WHO, neither are any legislation/guidelines available for monitoring and data collection	List of critical AMs to be notified. Guidelines to be formulated and legislation is required for monitoring AMs used.	AHC and FDC	DRAP, Customs, Prov L&DD, Prov DRA	Short term
Type of Data to be Provided to MoNFS&R and MoMA	No systematic mechanism is available; no legislation is available.	System to be developed for data collection and sharing with MoNFS&R and MoMA through AHC and FDC from the private sector as well as federal and provincial regulators	AHC, FDC	DRAP, Customs, Prov LDD, Prov DRA, Pharmaceuticals	Long term
Gaps and Barriers for AMU Data Collection	No guidelines, legislation, No SOPs, No system exists	Develop system, guidelines, SOPs, legislative reforms	AHC/FDC	DRAP, Customs, Prov L&DD, Prov DRA	Medium term
Way Forward	Political commitment, HR issues, legislative issues, need to develop a harmonized system for data collection, effective communication between federal and provinces, advocacy, and awareness of AMU.	Form a taskforce or technical working group (TWG) on AMU			

**Table 2 vetsci-12-00215-t002:** Initiatives to address gaps in monitoring of AMU at end-user level.

Topic	Current Gaps/Challenges	Way Forward	Lead Implementer	Collaborator	Timeline
AMU Monitoring	No system in place	Surveillance plan/legislation	Federal and Provinces	Public and Private	Medium term
Guidelines/SOPs	Don’t exist	Develop guidelines/SOPs	Federal and Provinces	Academia, Livestock Dept, Res. Institute, Association, Fish Dept	Medium term
Development of a monitoring system	No data base	Pilot study in different production systems	Federal and Provinces	Livestock Dept, Association Farmers	Short term
Publications	Only one that is too limited to the human health component [11]	More planned studies	Academia, Res. Institute	Livestock Dept, Association Farmers	Long term
Implementations and Challenges	Lack of awareness, trained manpower, funding, political interest	Allocation of budget, political will, training	Federal and Provinces	Planning and Finance	Long term
Roles and responsibilities	Federal legislation	Legislation, implementation, data analysis, reporting			
Knowledge, Attitudes, and Practices surveys	No planned activity	Planned study, multi-sectorial approach, awareness, AMR day	livestock dept	Academia, Research Institutes, NGO’s, Associations, Farmers	

## Data Availability

Data are contained within this paper.

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
