# Peer review of "Evaluation of Antimicrobial Usage Supply Chain and Monitoring in the Livestock and Poultry Sector of Pakistan"

_vetsci, 2025, doi:10.3390/vetsci12030215_

Round 1

Reviewer 1 Report

Comments and Suggestions for Authors

In general terms, I think the presented work is an evaluation/review of a specific issue in a specific country and it is good. I have certain doubts about whether this content should be published in "brief report" format, following the parameters Introduction/Materials and Methods/Results and Discussion...

However, a major point to consider for improving the work is to update the bibliographic references... the references used are minimal in number. There are paragraphs and paragraphs without a single reference... if the content is a finding of information from the Pakistani ministerial system, there must be websites, decrees, informative books, notices, etc., where this information is found. It is MANDATORY to cite and have a reference in the text.

The context of the information about Pakistan should be equally compared with situations in other countries with the same levels of development... I think the "discussion" should be improved at this level and in a comparative manner ALWAYS.

Good luck!

Author Response

Comment 1: “In general terms, I think the presented work is an evaluation/review of a specific issue in a specific country and it is good. I have certain doubts about whether this content should be published in "brief report" format, following the parameters Introduction/Materials and Methods/Results and Discussion…”

Response 1: We will request the respected editors to make a judgement on the type of Publication. However, we have aligned the paper to a brief report requirements by limiting the number of figures and tables. We have also changed the type of paper to “Project Brief” as suggested by another reviewer.

Comment 2: “However, a major point to consider for improving the work is to update the bibliographic references... the references used are minimal in number. There are paragraphs and paragraphs without a single reference... if the content is a finding of information from the Pakistani ministerial system, there must be websites, decrees, informative books, notices, etc., where this information is found. It is MANDATORY to cite and have a reference in the text.”

Response 2: Unfortunately, there has not been much work done in Pakistan on AMU, especially the supply chain system. The only paper available is now cited. The overall AMR work in Pakistan got pace after the launch of the Fleming Fund country grant, Pakistan and most of the authors are/were affiliated with the program hence limited livestock sector AMR papers are cited to reduce self-citations.

Comment 3: “The context of the information about Pakistan should be equally compared with situations in other countries with the same levels of development... I think the "discussion" should be improved at this level and in a comparative manner ALWAYS.”

Response 3: Added a few more comparisons, the supply chain evaluation from Lao PDR for example is now cited.

Reviewer 2 Report

Comments and Suggestions for Authors

The Authors present the recommendations resulted from a wotkshop that in 2020 gathered different stakeholders, mostly institutional, with the aim of defining antimicrobial supply chains in livestock and poultry sector in Pakistan. The article is interesting and offers some important guidelines for developing a monitoring system in the country to cope with the antibiotic resistance emergency, considered worldwide as a silent pandemic. The main difficulty I encountered in reading is the time-frame of the various passages which makes the work strategy rather confusing: in lines 116-117 the purpose of the work is the future assistance in designing and initiating a monitoring surveillance system as if nothing seems had been done at the moment the article is written, while in lines 340-341 it is stated that PPS surveys have been later carried out (in 2021 based on bibliographical references) for broilers and dairy in Pakistan. Given this fundamental aspect, I suggest first of all a full review of the work where both the outputs of the congress and the adopted PPS are taken into account in the results, summarising them and thus providing a general picture of AMU/AMR monitoring in the country over the 2020-2021 period; the review should also maintain a more concise approach, combining results and discussion in a single chapter to avoid the many repetitions (to give one example among many, a lack of AMU monitoring system is a statement repeated several times, being in fact both an outcome of the workshop and his solution a future expectation, as specified several times in the introduction, discussion and elsewhere);  the need for a greater synthesis is supported by the instructions for Authors: Brief Reports admit only 1 Figure and max 2 tables (so evaluate to reduce strongly the number of the tables in the manuscript).

Following some specific comments:

Lines 53-62: the Authors refer several times to the need to determine the causes of AMR; in fact the causes of AMR are universally known (in summary, overuse of antimicrobials with a selective pressure that determines resistant bacterial clones, antibiotic resistant gene transfer, etc); therefore I suggest rather than to talk about “investing the causes of AMR”, using for example this type of statement “the research of factors that favor the occurrence of AMR in Pakistan…”(mostly, as it's known, related to human practices as veterinary therapies without AMR test, growth promoters use, etc).

Line 66: insert also the full name of GDP.

Line 75: I suggest to change in “…both human and animals” among which really AMR increases (and not medicine).

Line 88: after “…and poultry)” probably lack a verb as “creating two National Reference lab…”.

Line 118: before paragraph 1.1 title of chapter is missing: “Materials and Methods”.

Line 119: After the first citation in line 94-95, use only the acronym for Ministry of National Food etc. (observe this rule through all the text for other acronyms involved).

Line 126: engaged

Table 1: I suggest deleting this Table (see also the comment above); if you consider keeping it, add for each organization/department the acronym (ex. WOAH, etc) and use only it in the text.

Line 142: I suggest describing here briefly the significance of the topics chosen (gaps, way forward, etc).

Figure 1: this Figure is – I thing rightly – referred in line 184 among the Results and also here among Materials and Methods. It’s not clear the time frame of this Figure, if provided to participants before the group work or if it's a final outcome of the congress.

In my opinion Figure 1 and the Table 3-5 are Results and so to be moved from Materials and Methods (the sequence however is not correct: Table 3 would be Table 2 after Table 1 with the list of participants to the workshop).

Tables: many acronyms are without definition (TWC, TORs, etc); check all these and add, when it’s the first time you cite, the name in full followed by the acronym.

Page 9, Table 5: cite the only one existing publication.

Line 176: antimonial, probably antimicrobial.

Line 232: use only ADI and MRL (see comments above), already defined in Table 3.

Paragraph 2.4: in line with the comments in introduction, this part is quite redundant with many concepts already expressed many times in the text. I suggest eliminating it.

Line 318: I suggest explaining the difference between AMC (antimicrobial consumption) and AMU (antimicrobial usage), as they look express the same concept.

Line 341: describe briefly the PPS survey sheets, considering that these seems the first practical application of the guidelines of the workshop in Islamabad.

Line 359: I suggest to change the title in “Conclusions”.

Author Response

Comment 1: “The Authors present the recommendations resulted from a wotkshop that in 2020 gathered different stakeholders, mostly institutional, with the aim of defining antimicrobial supply chains in livestock and poultry sector in Pakistan. The article is interesting and offers some important guidelines for developing a monitoring system in the country to cope with the antibiotic resistance emergency, considered worldwide as a silent pandemic. The main difficulty I encountered in reading is the time-frame of the various passages which makes the work strategy rather confusing: in lines 116-117 the purpose of the work is the future assistance in designing and initiating a monitoring surveillance system as if nothing seems had been done at the moment the article is written, while in lines 340-341 it is stated that PPS surveys have been later carried out (in 2021 based on bibliographical references) for broilers and dairy in Pakistan. Given this fundamental aspect, I suggest first of all a full review of the work where both the outputs of the congress and the adopted PPS are taken into account in the results, summarising them and thus providing a general picture of AMU/AMR monitoring in the country over the 2020-2021 period; the review should also maintain a more concise approach, combining results and discussion in a single chapter to avoid the many repetitions (to give one example among many, a lack of AMU monitoring system is a statement repeated several times, being in fact both an outcome of the workshop and his solution a future expectation, as specified several times in the introduction, discussion and elsewhere);  the need for a greater synthesis is supported by the instructions for Authors: Brief Reports admit only 1 Figure and max 2 tables (so evaluate to reduce strongly the number of the tables in the manuscript).”

Response 1: The major challenge for the establishment of a robust AMU surveillance in the livestock sector of Pakistan had been lack of knowledge on sources of antibiotics supply. This workshop was aimed at addressing this which subsequently led to some pilot surveys carried out with the support of the Fleming Fund Country Grant. However, there is still no routine AMU surveillance mechanism in the country.

As per guidance we have limited the tables and figures as suggested, now the paper has only one figure and two tables.

Following some specific comments:

Comment 2: “Lines 53-62: the Authors refer several times to the need to determine the causes of AMR; in fact the causes of AMR are universally known (in summary, overuse of antimicrobials with a selective pressure that determines resistant bacterial clones, antibiotic resistant gene transfer, etc); therefore I suggest rather than to talk about “investing the causes of AMR”, using for example this type of statement “the research of factors that favor the occurrence of AMR in Pakistan…”(mostly, as it's known, related to human practices as veterinary therapies without AMR test, growth promoters use, etc).”

Response 2: While the overall causes of AMR development are well known there is limited data on various associated factors from the low and middle income countries. Rephrased the sentence for clarity and better flow.

Comment 3: “Line 66: insert also the full name of GDP.”

Response 3: Added

Comment 4: “Line 75: I suggest to change in “…both human and animals” among which really AMR increases (and not medicine).”

Response 4: Modified the sentence accordingly

Comment 5: “Line 88: after “…and poultry)” probably lack a verb as “creating two National Reference lab…”.”

Response 5: Modified the sentence accordingly

Comment 6: “Line 118: before paragraph 1.1 title of chapter is missing: “Materials and Methods”.”

Response 6: Added the section

Comment 7: “Line 119: After the first citation in line 94-95, use only the acronym for Ministry of National Food etc. (observe this rule through all the text for other acronyms involved).”

Response 7: Changed the complete name to the acronym, apart from a couple of places where organizations are listed.

Comment 8: “Line 126: engaged”

Response 8: Corrected

Comment 9: “Table 1: I suggest deleting this Table (see also the comment above); if you consider keeping it, add for each organization/department the acronym (ex. WOAH, etc) and use only it in the text.”

Response 9: Deleted the table as suggested, the contents on the table are described in the section 2.3

Comment 10: “Line 142: I suggest describing here briefly the significance of the topics chosen (gaps, way forward, etc).”

Response 10: Modified the sentence for better clarity and flow.

Comment 11: “Figure 1: this Figure is – I thing rightly – referred in line 184 among the Results and also here among Materials and Methods. It’s not clear the time frame of this Figure, if provided to participants before the group work or if it's a final outcome of the congress.

In my opinion Figure 1 and the Table 3-5 are Results and so to be moved from Materials and Methods (the sequence however is not correct: Table 3 would be Table 2 after Table 1 with the list of participants to the workshop).”

Response 11: The figure is the outcome / result and hence has been moved to the results section as suggested. Table 1 and 3 has been deleted and their contents have been described in their relevant sections. Sequences of the tables are corrected. Table 4 & 5, now numbered as Table 1 & 2 are moved to the results section.

Comment 12: “Tables: many acronyms are without definition (TWC, TORs, etc); check all these and add, when it’s the first time you cite, the name in full followed by the acronym.”

Response 12: The full form of TWG has been added, the word TOR has been removed.

Comment 13: “Page 9, Table 5: cite the only one existing publication.”

Response 13: Cited, Ref [10]

Comment 14: “Line 176: antimonial, probably antimicrobial.”

Response 14: Corrected the typing error.

Comment 15: “Line 232: use only ADI and MRL (see comments above), already defined in Table 3.”

Response 15: Table 3 is now removed, text adjusted to make sure that full form are not repeated again and again and acronyms are used after the first instance.

Comment 16: “Paragraph 2.4: in line with the comments in introduction, this part is quite redundant with many concepts already expressed many times in the text. I suggest eliminating it.”

Response 16: It is basically the summary of findings, we suggest retaining it as it provides a concise overview of the results.

Comment 17: “Line 318: I suggest explaining the difference between AMC (antimicrobial consumption) and AMU (antimicrobial usage), as they look express the same concept.”

Response 17: Rather removed the term AMC to avoid any confusion

Comment 18: “Line 341: describe briefly the PPS survey sheets, considering that these seems the first practical application of the guidelines of the workshop in Islamabad.”

Response 18: Briefly explained in lines 379-381

Comment 19: “Line 359: I suggest to change the title in “Conclusions”.”

Response 19: Changed

Reviewer 3 Report

Comments and Suggestions for Authors

Only a few grammatical or typographical errors:

Line 85: country’s —> the country’s

In Table 3: Withdrawl —> Withdrawal

Line 260: lack —>lacks 

Line 72, 283, 284: Please consistently use attitude or attitudes

Line 322: had been —> has been

Author Response

Comment 1: “Line 85: country’s —> the country’s”

Response 1: Addressed

Comment 2: “In Table 3: Withdrawl —> Withdrawal”

Response 2: Addressed

Comment 3: “Line 260: lack —>lacks”

Response 3: Addressed

Comment 4: “Line 72, 283, 284: Please consistently use attitude or attitudes”

Response 4: Addressed

Comment 5: “Line 322: had been —> has been”

Response 5: Modified the sentence for clarity

Reviewer 4 Report

Comments and Suggestions for Authors

The authors pointed out that in Pakistan there is a lack of centralized database for monitoring the use of antibiotics in the livestock and poultry sector. It is encouraging that scientists in Pakistan want to introduce restrictions on the use of antibiotics in livestock as a promoter of animal growth in livestock. It should be noted that Ciprofloxacin and colistin are banded for use as antimicrobial growth promotors in poultry feed.

In the article, they presented how to establish a system for monitoring the use of antibiotics and a comprehensive surveillance plan and legislation. The authors want to build an integrated national system and were inspired by systems found in Europe and the United States. It is very important for veterinary in Pakistan, but in my opinion this article does not match the profile of this magazine.

Comments on the Quality of English Language

The flow of manuscripts is not very good.

Author Response

Comment 1: “The authors pointed out that in Pakistan there is a lack of centralized database for monitoring the use of antibiotics in the livestock and poultry sector. It is encouraging that scientists in Pakistan want to introduce restrictions on the use of antibiotics in livestock as a promoter of animal growth in livestock. It should be noted that Ciprofloxacin and colistin are banded for use as antimicrobial growth promotors in poultry feed.

In the article, they presented how to establish a system for monitoring the use of antibiotics and a comprehensive surveillance plan and legislation. The authors want to build an integrated national system and were inspired by systems found in Europe and the United States. It is very important for veterinary in Pakistan, but in my opinion this article does not match the profile of this magazine.”

Response 1: It is obviously up to the respected editors to determine the scope however in our humble opinion AMU supply chain monitoring in the livestock sector is one of the key interventions to tackle AMR in food animals, food supply chain and subsequently in humans. The authors have made an effort to map out the current supply chain, identify its gaps with the help of relevant experts and put forward some suggestions to improve it. These suggestions are based on the expert opinion of participants. The LMICs have their own challenges when it comes to tackling AMR and the authors have tried to present as local and tailored to the national situation solution as possible.

In our opinion AMR is a key One Health challenge globally and a major food safety concern therefore it aligns with the Food Safety and Zoonosis section of the journal especially the Journal’s Special issue: Occurrence and Antimicrobial Resistance of Bacterial Pathogens in Primary Animal Food Production. Moreover, we believe that our study can serve as a model for other LMICs facing similar challenges, the objective that has already been discussed in the manuscript and acknowledged by reviewer 2. Additionally, the relevance of the topic is evident by the fact that the preprint of this article has already been downloaded by 59 people in a short period.

Comments on the Quality of English Language

Comment 2: “The flow of manuscripts is not very good.”

Response 2: A major responsibility on the authors while publishing this paper has been to stay as transparent as possible to present the opinions from the participants, however keeping in view the comments of respected reviewers we have tried our best to improve the flow of manuscript.

Round 2

Reviewer 2 Report

Comments and Suggestions for Authors

Thank to the Authors for accepting almost all suggestions given in the first report. Pay attention to some acronyms that still require full names (consider that many readers in  the world may not know the meaning of these); check the test too if I missed anything. 

Table 1: NGO? HDPs and HDPsM? DRA?

Line 298: SOP's?

Line 349: PDR? Verify from the cited reference

Line 376: LMICs? Verify from the cited reference

Author Response

Comment:

Thank to the Authors for accepting almost all suggestions given in the first report. Pay attention to some acronyms that still require full names (consider that many readers in the world may not know the meaning of these); check the test too if I missed anything. 

Response: Thank you for the acknowledgement, we really appreciate your thorough review and are thankful for the valuable suggestions. We have now screened for all mentioned acronyms, removed the ones which were only mentioned once or twice and gave their full forms, kept the ones repeating more often and mentioned their full forms.

Comment: Table 1: NGO? HDPs and HDPsM? DRA?

Response: Added the full form of NGO, DRA mentioned while the other two acronyms removed.

Comment: Line 298: SOP's?

Response: Made the acronym consistent throughout the paper, now it is mentioned as “SOPs” while its full form is mentioned under heading 3.4, it is used before that in a table but refrained from mentioning in full to limit the text in the table.

Comment: Line 349: PDR? Verify from the cited reference

Response: Thank you for pointing out the acronym of Lao People’s Democratic Republic, removed this abbreviation altogether, Lao shall be sufficient in our opinion with the reference given right next.

Comment: Line 376: LMICs? Verify from the cited reference

Response: Mentioned the full form of LMICs

Reviewer 4 Report

Comments and Suggestions for Authors

I support my opinion that this article does not correspond to the profile of this magazine. I agree with the authors that it is necessary to improve regulations to restrict the use of antimicrobial growth promoters, establish an integrated national AMU database system, and raise awareness of the responsible use of antimicrobials in Pakistan's livestock and poultry sectors. However, this is within the rules of this country.

Comments on the Quality of English Language

The flow of manuscripts is not very good.

Author Response

Comment: I support my opinion that this article does not correspond to the profile of this magazine. I agree with the authors that it is necessary to improve regulations to restrict the use of antimicrobial growth promoters, establish an integrated national AMU database system, and raise awareness of the responsible use of antimicrobials in Pakistan's livestock and poultry sectors. However, this is within the rules of this country.

Response: Thank you for very thoughtful comments, regarding the profile we submitted the paper considering its relevance with the Journal’s Special issue: Occurrence and Antimicrobial Resistance of Bacterial Pathogens in Primary Animal Food Production. In our opinion AMR is a key One Health challenge globally and a major food safety concern therefore it aligns with the Food Safety and Zoonosis section of the journal.

Additionally, regarding the study design we have tried to explain the process a little more.

Comment: Comments on the Quality of English Language

The flow of manuscripts is not very good.

Response: We really appreciate your time and effort to review this manuscript, we have made another effort to improve the flow as per guidance from the reviewers. We hope the current version after second round of revision will meet your expectations.